

# Optimization of a tri-drug treatment against lung cancer using orthogonal design in preclinical studies

Jing Tan[1,*], Lijun Wang[1,*], Xuming Song[2], Yijian Zhang[2], Zhenghuan Song[1] and Manlin Duan[3]

[1] Department of Anesthesiology, Jiangsu Cancer Hospital & Jiangsu Institute of Cancer Research & The Affiliated Cancer Hospital of Nanjing Medical University, Nanjing, China
[2] Department of Thoracic Surgery, Jiangsu Key Laboratory of Molecular and Translational Cancer Research, Jiangsu Cancer Hospital & Nanjing Medical University Affiliated Cancer Hospital & Jiangsu Institute of Cancer Research, Nanjing, China
[3] Department of Anesthesiology, Jinling College Affiliated to Nanjing Medical University, Nanjing, China
* These authors contributed equally to this work.

Corresponding authors
Zhenghuan Song,
songzhenghuan@njmu.edu.cn
Manlin Duan, dml1200@126.com

## ABSTRACT

A growing body of evidence suggests that anesthetics impact the outcome of patients with cancer after surgical intervention. However, the optimal dose and underlying mechanisms of co-administered anesthetics in lung tumor therapy have been poorly studied. Here, we aimed to investigate the role of combined anesthetics propofol, sufentanil, and rocuronium in treating lung cancer using an orthogonal experimental design and to explore the optimal combination of anesthetics. First, we evaluated the effects of the three anesthetics on the proliferation and invasion of A-549 cells using Cell Counting Kit 8 and Transwell migration and invasion assays. Subsequently, we applied the orthogonal experimental design (OED) method to screen the appropriate concentrations of the combined anesthetics with the most effective antitumor activity. We found that all three agents inhibited the proliferation of A-549 cells in a dose- and time-dependent manner when applied individually or in combination, with the highest differences in the magnitude of inhibition occurring 24 h after combined drug exposure. The optimal combination of the three anesthetics that achieved the strongest reduction in cell viability was 1.4 μmol/L propofol, 2 nmol/L sufentanil, and 7.83 μmol/L rocuronium. This optimal 3-drug combination produced a more beneficial result at 24 h than either single drug. Our results provide a theoretical basis for improving the efficacy of lung tumor treatment and optimizing anesthetic strategies.

## INTRODUCTION

Lung cancer is the leading cause of cancer-related mortality worldwide (*Naccache et al., 2018*; *Wang et al., 2019*), with combined surgical resection and targeted therapy, immunotherapy, chemotherapy, or radiotherapy being the primary approaches for treating the disease. Although surgical intervention is the most effective method to improve patient

prognosis, the long-term clinical outcome of lung cancer is still unoptimistic, with five-year survival rates under 15% (*He et al., 2020*).

Perioperative anesthesia management of patients undergoing surgical procedures is essential for good surgical outcomes. Propofol, sufentanil, and rocuronium are common anesthetic medications administered during surgical interventions, often in combination, and have recently attracted much scientific attention owing to their antitumor effects.

Each has specific clinical significance in surgical intervention and has shown antitumor properties in preclinical studies. Propofol is an intravenous sedative-hypnotic drug used to induce and maintain anesthesia due to its rapid onset and offset of action, making it an ideal medication for surgical procedures. *Yu et al. (2019)* investigated the antitumor properties of propofol in preclinical studies and demonstrated it inhibits proliferation, migration, and invasion of human melanoma cells by regulating microRNA-137 and fibroblast growth factor 9 (FGF9). *Zhang et al. (2020)* uncovered that propofol inhibits invasion and promotes apoptosis of colon cancer cells by preventing the interaction between signal transducer and activator of transcription 3 (STAT3) and the HOX anti-sense intergenic RNA (HOTAIR) promoter, suppressing the Wnt pathway *via* WIF1. Similarly, *Sun et al. (2021)* showed propofol inhibits cervical cancer progression by controlling the HOTAIR/miR-129-5p/RPL14 axis. Sufentanil is a potent opioid analgesic often used as an adjunct to general anesthesia. Its clinical significance lies in its ability to provide efficient analgesia during surgery and minimize the risk of adverse effects associated with opioid administration. *Wu et al. (2014)* studied its antitumor effects and found it suppresses the viability of gastric cancer SGC-7901 cells and induces their apoptosis *in vitro*. *Guan, Huang & Lin (2022)* demonstrated that sufentanil regulates the Wnt pathway to inhibit various functions of lung cancer cells, including proliferation, migration, invasion, and epithelial-mesenchymal transition.

Rocuronium is a nondepolarizing neuromuscular blocking agent that facilitates endotracheal intubation and muscle relaxation during an operation, allowing easy and safe surgical access. *Jiang et al. (2016)* evidenced that rocuronium bromide promotes invasion, adhesion, and growth of breast cancer MDA-231 cells.

A combination of rocuronium and propofol can provide excellent intubating conditions, reducing the discomfort of propofol administration. Combining opioids with rocuronium is also beneficial for intubation conditions and attenuates pain during administration (*Costa, Mourão & Vale, 2022*). Moreover, the individual antitumor effects of propofol, sufentanil, and rocuronium make them potential candidates for combination therapy in the surgical management of lung cancer. Although the combined administration of these drugs in lung cancer surgical intervention has gained considerable attention due to the clinical significance and antitumor effects of individual drugs, the optimal dose of co-administered anesthetics and the underlying mechanisms of the antitumor effects associated with drug co-administration against lung cancer are unclear. We hypothesized that these drugs each have a multi-targeted action, inhibiting cancer cell proliferation, inducing apoptosis, and suppressing tumor growth and invasion. Therefore, synergistic antitumor effects of propofol, sufentanil, and rocuronium may provide a more effective treatment approach for patients with lung cancer and improve the disease outcome.

Orthogonal tests are increasingly used in experimental studies to investigate the influences of multiple factors and levels on test results with orthogonal tables (*Wu et al., 2021*). These tests allow choosing of representative combinations for experimentation, whose respective impacts are analyzed to determine their optimal combination, offering a cost-effective, fast, and economic strategy for experiment design (*Feng et al., 2019*; *Gao et al., 2015*; *Muheem et al., 2017*; *Zhao et al., 2017*). Here, we present an analysis of the factors affecting the proliferation of lung cancer A-549 cells based on an orthogonal experimental design. We identified propofol and sufentanil as critical factors for A-549 cell viability and confirmed the optimal combination treatments that exerted synergistic reductions in viability. These comprehensive assessments could provide the basis for identifying appropriate types and combinations of therapeutic targets, which may improve patient prognosis.

## MATERIALS AND METHODS

### Biological reagents and cell cultures

Lung cancer A-549 cells were purchased from and authenticated at the Institute of Biochemistry and Cell Biology, Shanghai Institutes for Biological Sciences, Chinese Academy of Sciences (Shanghai, China). The cells were grown in Dulbecco's Modified Eagle Medium (DMEM) medium (Jiangsu KeyGEN BioTECH Co., Ltd., Nanjing, China, KGM12800N) containing 10% FBS (#10099141; Gibco, MA, USA) and 1% penicillin-streptomycin (#15140122; Gibco, MA, USA). All cell cultures were incubated at 37 °C in a humidified atmosphere incubator with 5% $CO_2$. The anesthetics were purchased as follows: propofol, Selleck Chemicals (Huston, TX, USA); sufentanil, Jiangsu Enhua Pharmaceutical Co., Ltd (China); and rocuronium, Zhejiang Xianju Pharmaceutical Co., Ltd (China).

### Drug treatment

We selected the median, 1/3 of the median, and 3× the median of the concentration ranges suggested by a Chinese consensus for intravenous anesthesia (*Yun, 2016*). For the cell viability assay, A-549 cells were incubated with propofol (1.4, 4.2, and 12.6 μmol/L), sufentanil (0.67, 2, and 6 nmol/L), and rocuronium (2.61, 7.83, and 23.5 μmol/L) for 12, 24, 36, and 48 h.

### Cell viability assay

The survival rate of A-549 cells was determined using a Cell Counting Kit 8 (CCK-8) (Jiangsu KeyGEN BioTECH Co., Ltd., Nanjing, China), following the manufacturer's instructions. The cells were seeded at $1 \times 10^5$ cells per well into 96-well plates containing propofol, sufentanil, rocuronium or a combination of the three anesthetics at the indicated concentrations and times. The CCK-8 solution was added to each well, followed by a 3-h incubation at 37 °C. The absorbance at 450 nm was determined with a microplate reader, and data were analyzed using SoftMax Pro software (SoftMax Pro software version 6.4; Molecular Devices LLC, San Jose, CA, USA). Three duplicates of all samples were analyzed.

**Table 1 Three levels of the three factors in the orthogonal design.** Three factors, namely propofol, sufentanil, and rocuronium. Each factor had three concentration levels. Propofol concentration levels were designated as Level 1 (1.4 µmol/L), Level 2 (4.2 µmol/L), and Level 3 (12.6 µmol/L). Sufentanil concentration levels were categorized as Level 1 (0.67 nmol/L), Level 2 (2 nmol/L), and Level 3 (6 nmol/L). Rocuronium concentration levels were denoted as Level 1 (2.61 µmol/L), Level 2 (7.83 µmol/L), and Level 3 (23.5 µmol/L).

| Factor | | Propofol (µmol/L) | Sufentanil (nmol/L) | Rocuronium (µmol/L) |
|---|---|---|---|---|
| | 1 | 1.4 | 0.67 | 2.61 |
| Level | 2 | 4.2 | 2 | 7.83 |
| | 3 | 12.6 | 6 | 23.5 |

**Table 2 Results of the orthogonal design evaluation method.**

| Experimental number | Factor | | | Mean ± SD | *P*-value |
|---|---|---|---|---|---|
| | Propofol | Sufentanil | Rocuronium | | |
| 1 | 1 | 1 | 1 | $0.751 \pm 0.124$ | $P = 0.0061$[**] |
| 2 | 1 | 2 | 3 | $0.751 \pm 0.124$ | $P = 0.0061$[**] |
| 3 | 1 | 3 | 2 | $0.680 \pm 0.119$ | $P = 0.0003$[**] |
| 4 | 2 | 1 | 3 | $0.712 \pm 0.069$ | $P = 0.0011$[**] |
| 5 | 2 | 2 | 2 | $0.728 \pm 0.165$ | $P = 0.0023$[**] |
| 6 | 2 | 3 | 1 | $0.587 \pm 0.121$ | $P < 0.0001$[**] |
| 7 | 3 | 1 | 2 | $0.681 \pm 0.023$ | $P = 0.0003$[**] |
| 8 | 3 | 2 | 1 | $0.701 \pm 0.227$ | $P = 0.0007$[**] |
| 9 | 3 | 3 | 3 | $0.538 \pm 0.051$ | $P < 0.0001$[**] |
| K1 | 2.18 | 2.14 | 2.04 | | |
| K2 | 2.03 | 2.18 | 2.09 | | |
| K3 | 1.92 | 1.8 | 2 | | |
| R | 0.09 | 0.13 | 0.03 | | |

**Notes.**

Propofol: 1, 2, and 3 represent 1.4, 4.2, and 12.6 µmol/L, respectively;

Sufentanil: 1, 2, and 3 represent 2/3, 2, and 6 nmol/L, respectively; rocuronium:1, 2, and 3 represent 2.61, 7.83, and 23.5 µmol/L, respectively.

*K1, K2,* and *K3* represent the mean scores for each factor with different levels, respectively. R indicates the range of average inhibition rates of various factors (range = maximum average inhibitory rate + minimum average inhibitory rate). Data were analyzed as mean ± standard deviation using the orthogonal experiment intuitionistic analytical method.

[*]$p < 0.05$.

[**]$p < 0.01$.

## Orthogonal experimental design (OED)

Our preliminary experimental results revealed that propofol, sufentanil, and rocuronium influence A-549 cell viability at different concentrations. Each drug constituted one factor, so three factors and three concentrations of each drug were evaluated. Each of the concentrations represented 1 level (Table 1). Thus, a 3-factor, 3-level L9($3^3$) OED was selected and applied to the experiments (Table 2) (*Manhas, Kumar & Chaubey, 2022*).

## Transswell assay

Serum-starved A-549 cells ($5 \times 10^4$) were suspended in 200 µL of serum-free DMEM medium containing 1.4 µmol/L propofol, 2 nmol/L sufentanil, and 7.83 µmol/L rocuronium. A mixture of serum free-medium and pretreated cells was seeded into the upper chamber of matrigel-coated inserts (Corning Inc., Corning, NY, USA). For the migration experiment, the upper chamber was coated with glue without a Matrigel matrix, while the lower chamber was filled with culture medium containing 10% FBS and incubated at 37 °C for 24 h. Non-migratory or non-invasive cells were gently removed from the upper chamber with a cotton bud, fixed, and stained using 4% paraformaldehyde and 0.1% crystal violet (#C0121; Beyotime Institute of Biotechnology). The invasive cells in 5 randomly selected visual fields were counted under a TS100-F inverted microscope (Nikon Corporation, Tokyo, Japan) at 200× magnification. The average number of cells passing through the membrane was calculated using ImageJ version 1.8.0.

## Wound-healing assay

The migration ability of A-549 cells was assessed with a scratch wound assay. In brief, the cells were seeded into 6-well plates at $5 \times 10^5$ cells/well and cultured to confluence. A gap was created in the cell monolayer by scratching it with a pipetting tip. The debris was removed with phosphate-buffered saline, and a serum-free medium was added. Micrographs were recorded at 0 and 24 h, and a migration distance analysis was performed using ImageJ software. The healing rate (%) was calculated as follows: (0 h scratch width $-24$ h scratch width)/0 h scratch width $\times 100\%$.

## Annexin V-fluorescein isothiocyanate (FITC)/PI double-labeling and flow cytometry

The cells were seeded into a 6-well plate at $2 \times 10^5$ cells/well and treated with optimal concentrations of the combined drugs. After 48 h, the cells were trypsinized, collected, and resuspended in PBS for cell counting. Cell concentration was adjusted to $5 \times 10^5$ cells/mL, and the cells were stained with Annexin V-FITC and PI (Kaiji Biotechnology Development, Co., Ltd., Nanjing, China) for 15 min at room temperature. Flow cytometry analysis was performed using a flow cytometer (BD Biosciences) with excitation at 380 nm and emission at 525 nm. Results were recorded and analyzed using CELLQuest software (BD Biosciences).

## Western blot analysis

Total protein was extracted from cells using Radio Immunoprecipitation Assay (RIPA) lysis buffer and quantified with a BCA Protein Assay Kit (Beyotime Biotechnology, Shanghai, China). The proteins were separated on a 10% sodium dodecyl sulfate and polyacrylamide gel by electrophoresis according to the molecular weights. The separated proteins were transferred onto polyvinylidene fluoride membranes (Millipore) using a semidry transfer system and immobilized for subsequent immunodetection. The membranes were blocked with 5% non-fat milk to prevent nonspecific binding and probed with primary antibodies overnight at 4 °C. The primary antibodies used were anti-PARP1 (ab191217; Abcam) at 1:1500 dilution or anti- $\beta$-actin (ab8227; Abcam) at 1:2500 dilution. The membranes were

washed with tris-buffered saline (TBST) and Tween 20 buffer, followed by incubation with a fluorescent-labeled DyLight 680 goat anti-mouse IgG secondary antibody (A23710, Abbkine Scientific Co., Ltd, Wuhan, China) at 1:20000 dilution. The secondary antibody was conjugated with a Horseradish Peroxidase (HRP) label, enabling primary antibody detection. The membranes were washed with TBST buffer to remove any unbound secondary antibody, and the protein signal was visualized and quantified using an Odyssey Fc Imager (LI-COR Biosciences, Lincoln, NE, USA).

## Statistical analysis

Data were expressed as mean $\pm$ SD and statistically analyzed using SPSS software version 13 (SPSS Inc., Chicago, IL, USA). The results were displayed as graphs using GraphPad Prism version 8.0.2 (GraphPad Software, Inc., San Diego, CA, USA). The differences between different treatment groups were assessed using a 1-way analysis of variance (ANOVA) corrected for multiple comparisons (Bonferroni test). Statistical significance was inferred when $P < 0.05$.

## RESULTS

We used the orthogonal experimental design (OED) to examine the effectiveness of a co-treatment with propofol, sufentanil, and rocuronium on the proliferation of lung adenocarcinoma A-549 cells. Because the traditional OED design involves 27 trials, the exhaustive examination of all possible combinations would demand an unreasonably high workload. Thus, using the L9 ($3^3$) orthogonal array, we obtained nine formulations with direct and variance analysis to determine the degree of influence of each anesthetic in the combined treatment (Table 2). Our data showed that the inhibition of cell proliferation significantly increased with treatment time when the cells were co-treated with the three drugs using different combinations of drug concentrations. Moreover, the differences in the magnitude of inhibition were most noticeable at 24 h, with later time points showing insignificant differences (Fig. 1). We also calculated a range for the test index under various factors and levels and revealed that sufentanil had the largest range (range, 0.13), followed by propofol (range, 0.09) and rocuronium (range, 0.03). (Tables 2 and 3). Finally, we determined the optimal concentrations of the three drugs for the co-treatment were 1.4 μmol/L propofol, 2 nmol/L sufentanil, and 7.83 μmol/L rocuronium. This optimal combination of the three drugs caused a stronger inhibition of A-549 cell proliferation than either treatment with an individual drug at 24 h, as illustrated in Fig. 2 ($P < 0.01$).

## Effect of propofol, sufentanil, and rocuronium on A-549 cell proliferation

The effect of propofol, sufentanil, and rocuronium on the proliferation of A-549 cells was evaluated using the CCK8 assay. Propofol treatment significantly inhibited the proliferation of A-549 cells compared with control cells. The inhibition of cell proliferation was observed at 3 different concentrations of the drug (1.4, 4.2, and 12.6 μmol/L) and at various time points (12, 24, 36 and 48 h) after the treatment. The proliferation of A-549 cells was inhibited under all propofol concentrations at 12 h posttreatment (control, 1.00 $\pm$ 0.08; 1.4

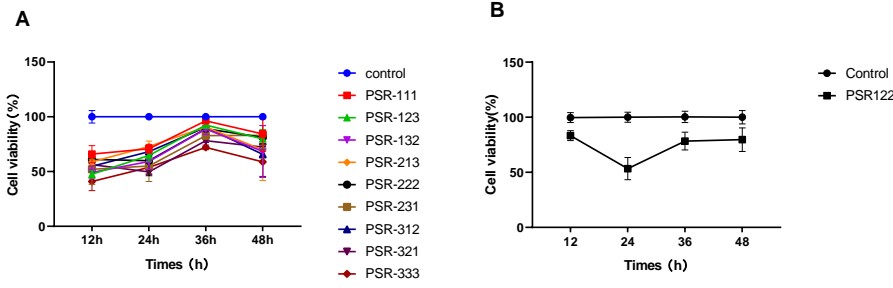

**Figure 1   Selection of optimal combinations of anesthetic agents using the orthogonal design experimental method.** (A) Viability of lung adenocarcinoma A-549 cells under the combined treatment with propofol, sufentanil, and rocuronium at different drug concentrations. The selected optimal parameter combination was 1.4 μmol/L propofol, 2 nmol/L sufentanil, and 7.83 μmol/L rocuronium. (B) Comparison of cell viability under the optimal combination of drugs and mock treatment (control). The inhibitory effect of various concentrations of the anesthetic agents on the proliferation of A-549 cells was assessed at 12 and 48 h posttreatment. The differences in the extent of inhibition were the most prominent at the 24-h time point. The data represent the mean ± SD from 3 independent experiments.P, propofol; 1, 1.4 μmol/L; 2, 4.2 μmol/L; and 3, 12.6 μmol/L.S, sufentanil; 1, 0.67 nmol/L; 2, 2 nmol/L; and 3, 6 nmol/L.R, rocuronium; 1, 2.61 μmol/L; 2, 7.83 μmol/L; and 3, 23.5 μmol/L.PSR 111 (propofol 1.4 μmol/L, sufentanil 0.67 nmol/L, and rocuronium 2.61 μmol/L), PSR123 (propofol 1.4 μmol/L, sufentanil 2 nmol/L, and rocuronium 23.5 μmol/L), PSR132 (propofol 1.4 μmol/L, sufentanil 6 nmol/L, and rocuronium 7.83 μmol/L), PSR213 (propofol 4.2 μmol/L, sufentanil 0.67 nmol/L, and rocuronium 23.5 μmol/L), PSR222 (propofol 4.2 μmol/L, sufentanil 2 nmol/L, and rocuronium 7.83 μmol/L), PSR231 (propofol 4.2 μmol/L, sufentanil 6 nmol/L, and rocuronium 2.61 μmol/L), PSR312 (propofol 12.6 μmol/L, sufentanil 0.67 nmol/L, and rocuronium 7.83 μmol/L),PSR321 (propofol 12.6 μmol/L, sufentanil 2 nmol/L, and rocuronium 2.61 μmol/L), PSR333 (propofol 12.6 μmol/L, sufentanil 6 nmol/L, and rocuronium 23.5 μmol/L), PSR122 (propofol 1.4 μmol/L, sufentanil 2 nmol/L, and rocuronium 7.83 μmol/L).

**Table 3   Analytical results of the variance of the co-combination drugs.**

| Source of variation | Mean-square | F | P |
|---|---|---|---|
| (Intercept) | 4.174 | 6674.599 | 0.000[**] |
| sufentanil | 0.014 | 22.818 | 0.042[*] |
| rocuronium | 0.001 | 1.038 | 0.491 |
| propofol | 0.006 | 9.25 | 0.098 |
| $R^2$: | 0.971 | | |

**Notes.**
The data were evaluated by ANOVA. Sufentanil was the most effective component in the reduction of A549 cell proliferation.
[*]$p < 0.05$.
[**]$p < 0.01$.

μmol/L propofol, 0.80 ± 0.12; 4.2 μmol/L propofol, 0.73 ± 0.09; 12.6 μmol/L propofol, 0.70 ± 0.08) (Fig. 3A), and a similar inhibitory effect was observed at 24 h (control, 1.00 ± 0.02; 1.4 μmol/L propofol, 0.78 ± 0.01; 4.2 μmol/L propofol, 0.68 ± 0.02; 12.6 μmol/L propofol, 0.63 ± 0.01) (Fig. 3B). The inhibitory effect was maintained at 36 h (control, 1.00 ± 0.04; 1.4 μmol/L propofol, 0.84 ± 0.03; 4.2 μmol/L propofol, 0.75 ± 0.05; 12.6 μmol/L propofol, 0.70 ± 0.04) (Fig. 3C) and 48 h posttreatment (control, 1.00 ± 0.02; 1.4 μmol/L propofol, 0.86 ± 0.01; 4.2 μmol/L propofol, 0.78 ± 0.05; 12.6 μmol/L propofol,

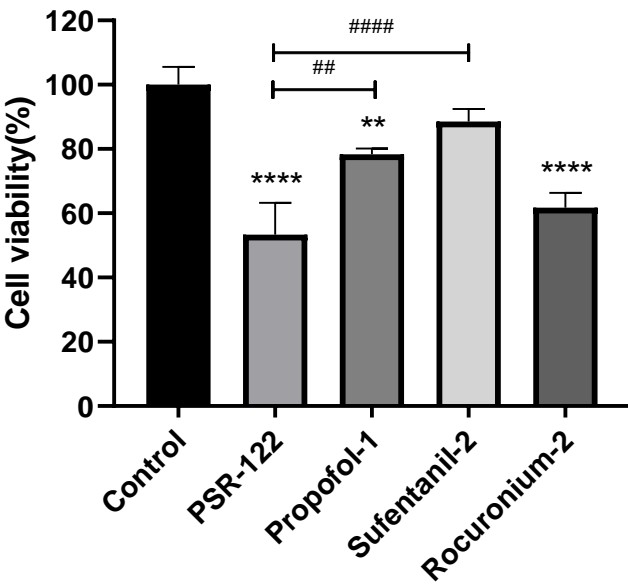

**Figure 2 Comparison of cell viability between single-agent treatments and treatments with the optimal combination of agents.** Significantly stronger inhibition of lung adenocarcinoma A-549 cell proliferation was achieved by the optimal combination of propofol, sufentanil, and rocuronium than either drug. The data represent the mean ± SD from 3 independent experiments. *$P < 0.05$, **$P < 0.01$, ***$P < 0.001$, ****$P < 0.0001$ *vs* control group; #$P < 0.05$, ##$P < 0.01$, ###$P < 0.001$, ####$P < 0.0001$ *vs* PSR122 group. PSR122 stands for PSR, propofol, sufentanil, and rocuronium; 1, 1.4 μmol/L propofol; 2, 2 nmol/L sufentanil; and 2, 7.83 μmol/L rocuronium.

$0.70 \pm 0.01$) (Fig. 3D). These results indicate that propofol inhibits A-549 cell proliferation for a prolonged time, suggesting an inhibitory effect of propofol on the growth of lung cancer cells.

Sufentanil inhibited the viability of A-549 cells but only at specific concentrations and time points. For instance, at 0.67 nmol/L concentrations, sufentanil had no effect on A-549 cell proliferation compared with control cells ($1.00 \pm 0.1$). By contrast, at 2 and 6 nmol/L, it significantly inhibited cell proliferation at 12, 24, 36 and 48 h (control group: $1.00 \pm 0.03$, $1.00 \pm 0.01$, $1.00 \pm 0.03$, and $1.00 \pm 0.01$; 2 nmol/L group: $0.57 \pm 0.01$, $0.59 \pm 0.02$, $0.79 \pm 0.05$, and $0.66 \pm 0.01$; and 6 nmol/L group: $0.86 \pm 0.06$, $0.89 \pm 0.04$, $0.71 \pm 0.01$, and $0.90 \pm 0.04$, as shown in Figs. 4A, 4B, and 4D). These results indicate that intermediate to high doses of sufentanil effectively suppress the growth of A-549 cells from 0 to 48 h, and low dose cannot exert this effect.

Rocuronium inhibited the proliferation of A-549 cells at 7.83 μmol/L and 23.5 μmol/L compared with the control cells (Fig. 5). The inhibitory effect at either drug concentration was apparent at 12 h (control, $1.00 \pm 0.01$; 7.83 μmol/L rocuronium, $0.62 \pm 0.06$; 23.5 μmol/L rocuronium, $0.48 \pm 0.02$) (Fig. 5A), 24 h (control, $1.01 \pm 0.06$; 7.83 μmol/L rocuronium, $0.62 \pm 0.01$; 23.5 μmol/L rocuronium, $0.54 \pm 0.01$) (Fig. 5B), 36 h (control, $1.00 \pm 0.05$; 7.83 μmol/L rocuronium, $0.65 \pm 0.05$; 23.5 μmol/L rocuronium, $0.52 \pm 0.03$) (Fig. 5C), and 48 h posttreatment (control, $1.00 \pm 0.02$; 7.83 μmol/L rocuronium, $0.69 \pm 0.03$; 23.5 μmol/L rocuronium, $0.62 \pm 0.02$) (Fig. 5D).

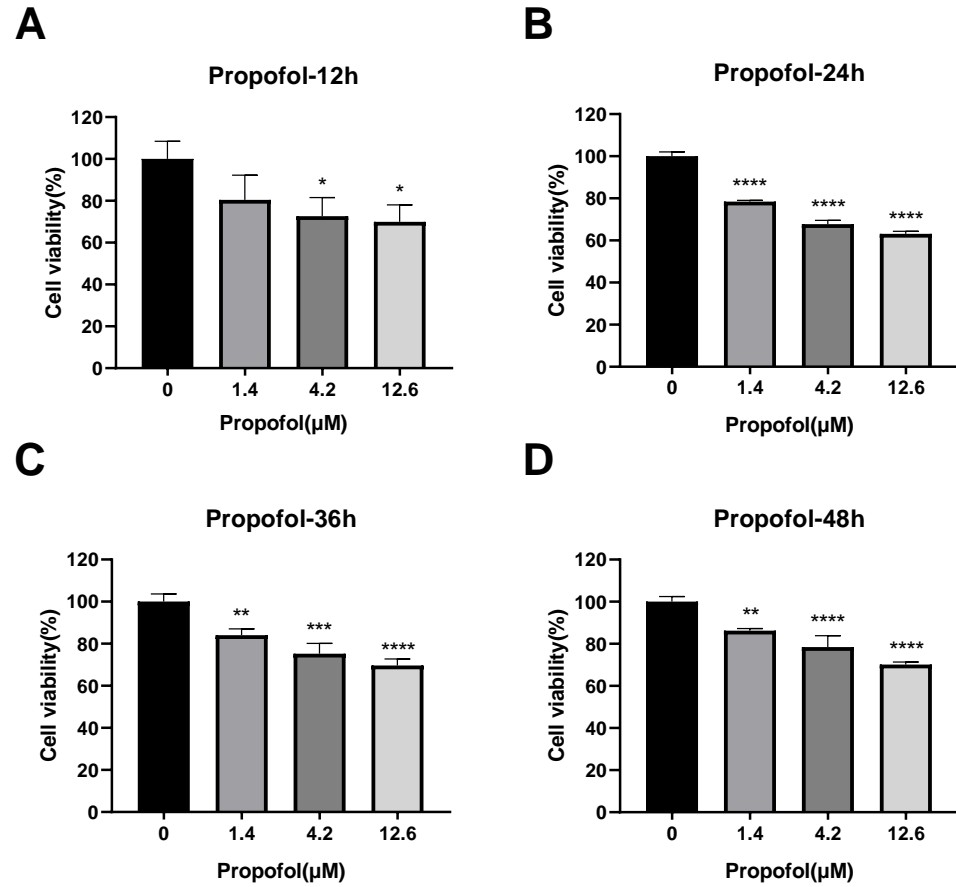

**Figure 3** **Effects of propofol treatment on the viability of lung adenocarcinoma A-549 cells at different drug concentrations and treatment times.** Cell viability was evaluated at 12 h (A), 24 h (B), 36 h (C), and 48 h posttreatment (D). The data represent the mean ± SD from 3 independent experiments. *$P < 0.05$, **$P < 0.01$, ***$P < 0.001$, ****$P < 0.0001$ *vs* control group.

The three-drug treatment inhibited the migration, invasion, and proliferation of the lung adenocarcinoma cell line, confirming the inhibitory effect of the combined drugs on lung cancer cells. Transwell and wound-healing assays were used to assess migration and invasion of A-549 cells co-treated with the three drugs and explore their antitumor effects. The results indicated that the invasion and migration of A-549 cells were markedly restrained following the co-treatment ($P < 0.05$) (Figs. 6A, 6B, 6C).

### The 3-drug combination induces apoptosis of A-549 cells

Because the three drugs repress cell proliferation, we evaluated whether they also influence apoptosis of A-549 cells using Annexin V FITC/PI staining and flow cytometry. We verified this result by investigating the expression of the apoptosis marker poly (ADP-ribose) polymerase 1 (PARP1) in A-549 cell extracts by Western blotting. We treated the cells with the three anesthetics at optimal concentrations and observed PARP1 expression after one to three half-lives of each drug. As shown in Fig. 6D, the PARP1 protein levels were significantly upregulated in the cells co-treated with the three drugs and gradually lowered

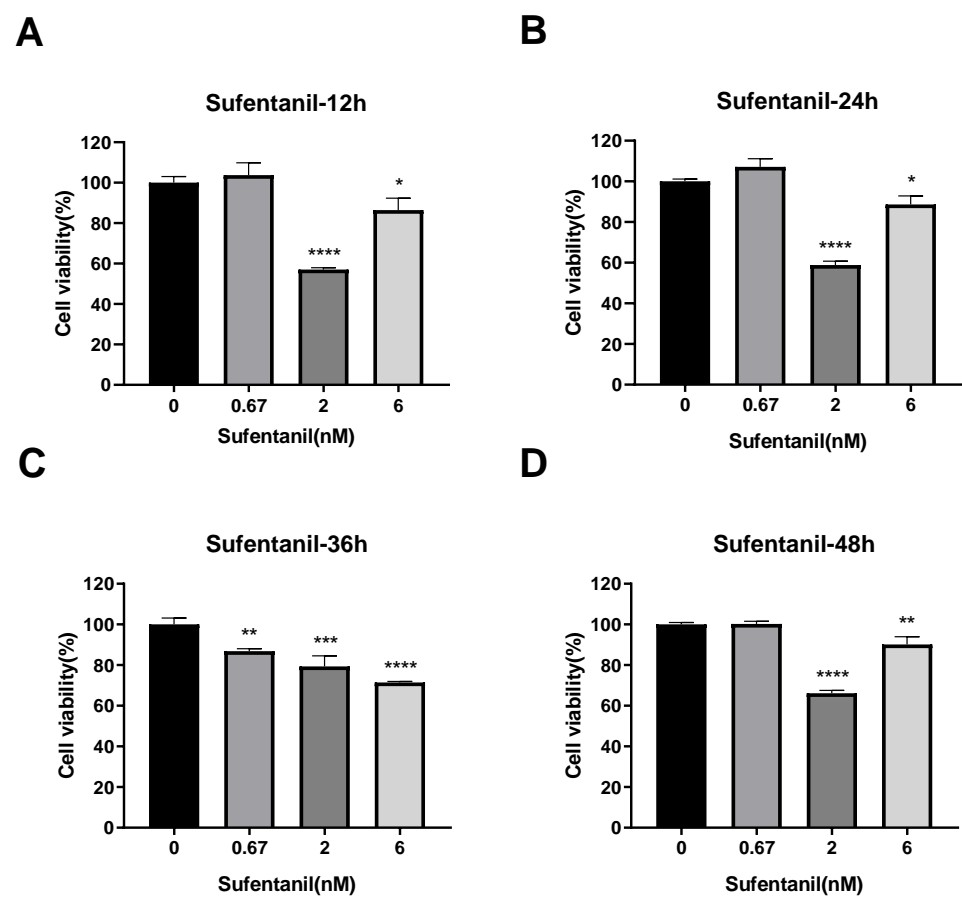

**Figure 4** **Effects of sufentanil treatment on the viability of lung adenocarcinoma A-549 cells at different drug concentrations and treatment times.** Cell viability was evaluated at 12 h (A),24 h (B), 36 h (C), and 48 h posttreatment (D). The data represent the mean ± SD from three independent experiments. *$P < 0.05$, **$P < 0.01$, ***$P < 0.001$, ****$P < 0.0001$ *vs* control group.

with the decrease in drug concentration (*i.e.,* with each consecutive half-life). The flow cytometry analysis showed that the drug-treated group significantly promoted apoptosis of A-549 cells (Fig. 6E). These results suggest that the optimal treatment may inhibit the ability of cell proliferation by promoting apoptosis.

## DISCUSSION

Despite the remarkable advances in lung cancer treatment, surgical resection has remained the most effective and potentially curative treatment modality. However, during the perioperative period, a surgical procedure may facilitate the entry of circulating tumor cells into the bloodstream (*Cristofanilli et al., 2004*; *Pachmann et al., 2005*; *Pelaz et al., 2017*), allowing them to disseminate the tumor at distant sites. In some patients with prostate cancer, for example, surgical resection rapidly reduces the circulating cells within 24 h posttreatment, while in others, these cells persist for months (*Stott et al., 2010*). Because circulating tumor cells have proliferative potential, finding an effective

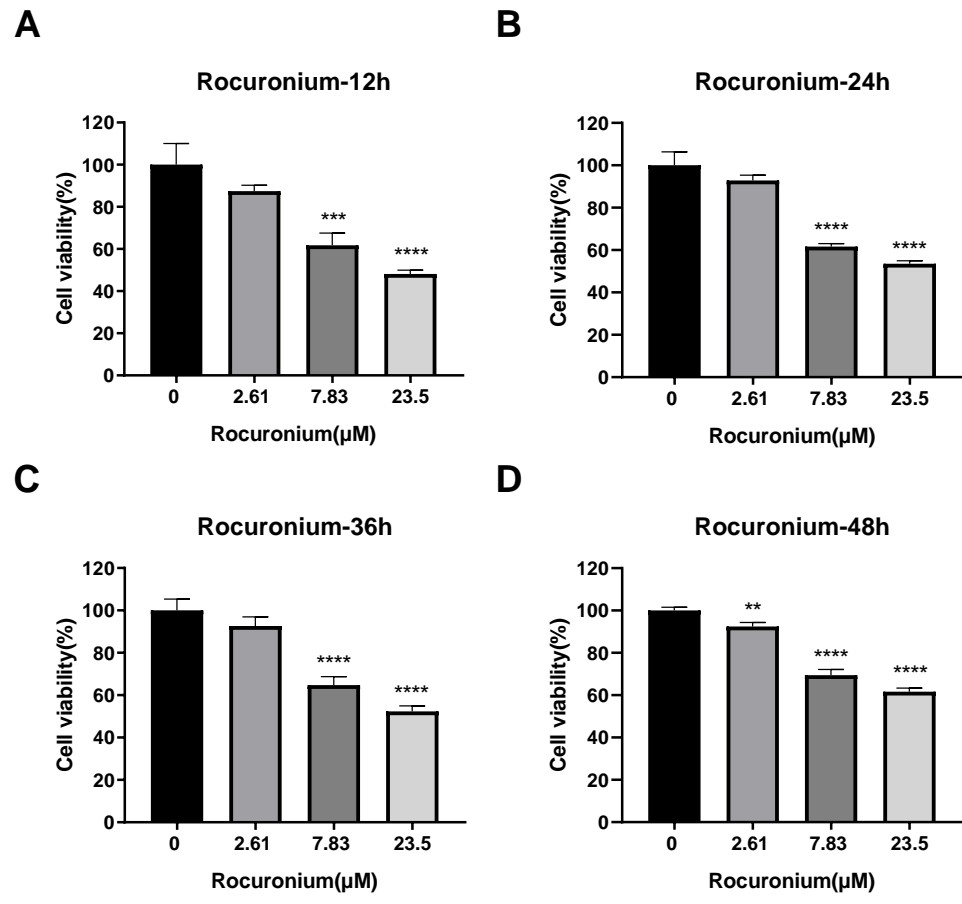

**Figure 5  Effects of rocuronium treatment on the viability of lung adenocarcinoma A-549 cells at different drug concentrations and treatment times.** (A) A-549 Cell viability was evaluated at 12 h (A),24 h (B), 36 h (C), and 48 h posttreatment (D). The data represent the mean ± SD from three independent experiments. *$P < 0.05$, **$P < 0.01$, ***$P < 0.001$, ****$P < 0.0001$ *vs* control group.

strategy for their elimination is essential for lowering the risk of tumor repopulation after surgical resection and improving the prognosis. Anesthesia is a crucial component of an operation and may impact tumor development and prognosis. Propofol, sufentanil, and rocuronium are the most utilized drugs in total intravenous anesthesia, with anesthetic and widespread non-anesthetic effects (*Bundscherer et al., 2015*; *Jiang et al., 2017*; *Tian et al., 2020*). Propofol, for instance, possesses antioxidant, immunomodulatory, and neuroprotective properties (*Vasileiou et al., 2009*). Sufentanil induces postoperative analgesia, which reduces postoperative pain, improves liver function following an operation, and alleviates immunosuppression in rats with hepatocellular carcinoma that underwent hepatectomy (*Peng et al., 2020*).

A growing body of research suggests that anesthetic drugs are associated with tumor suppression. Cancerous tumors are characterized by uncontrolled cell proliferation, and many anti-cancer drugs work by disrupting the process (*Mens & Ghanbari, 2018*; *Peng et al., 2020*). Thus, we devised a concentration gradient to test the inhibitory effects of

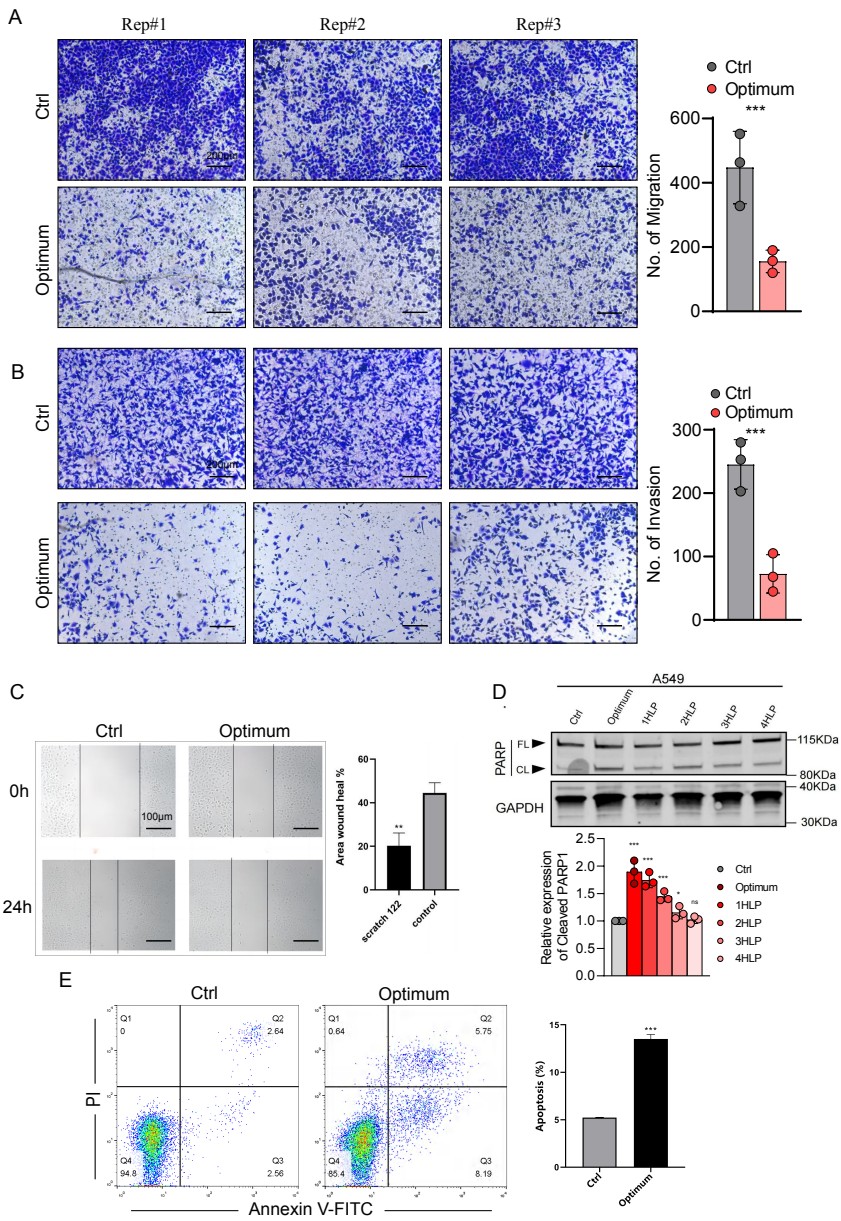

**Figure 6** **Effects of the optimal combination of three anesthetics on the migration and invasion of lung adenocarcinoma A-549 cells.** (A) Left, representative image showing the migration of A-549 cells in a Transwell assay. Right, quantitation of migrating A-549 cells. (B) Left, representative image depicting the invasion of A-549 cells in a Transwell assay. Right, quantitation of invasive A-549 cells. (C) Quantitation of migrating A-549 cells by wound-healing assay. (D) Top, poly (ADP-ribose) polymerase 1 (PARP1) protein levels in A-549 cells assessed by Western blotting. $\beta$-Actin served as a loading control. Bottom, quantitation of PARP1 cleavage in A-549 cells estimated by Western blotting. (E) Annexin V-FITC/PI double staining method for assessing apoptosis induced by a combination of anesthetic drugs in lung adenocarcinoma cell lines. Data were calculated from 3 independent experiments and expressed as the mean ± SD of the three experiments. $^*P < 0.05$, $^{**}P < 0.01$, $^{***}P < 0.001$, $^{****}P < 0.0001$ *vs* control group. Optimum; 1.4 µmol/L propofol, 2 nmol/L sufentanil, and 7.83 µmol/L rocuronium; 1 HLP, one half-life; 2 HLP, two half-lives; 3 HLP, three half-lives; 4 HLP, four half-lives.

propofol, sufentanil, and rocuronium on lung cancer A-549 cells. We found that each had a specific inhibitory effect on the proliferation of these cells. For example, propofol suppressed cell proliferation at various concentrations and treatment times, consistent with the findings of other investigations (*Huang, Lei & Liu, 2020*; *Liu & Liu, 2018*; *Sun & Gao, 2018*). Therefore, propofol may function as an ideal anesthetic agent in lung cancer surgical intervention due to its sustained inhibitory effect on cancer cell proliferation.

Evidence indicates that opioid-induced cell proliferation is concentration-dependent (*Wu et al., 2014*). While tumor growth is promoted by applying low or single doses of opioids, it is restricted under high or chronic opioid exposure (*Kumar et al., 2017*). We showed that medium (2 nmol/L) or high-dose sufentanil (6 nmol/L) inhibited cell proliferation, but low- (0.67 nmol/L) dose (2 nmol/L) did not, agreeing with the previous study (*Tegeder et al., 2003*). We believe the discrepancy between the effects observed after exposure to different doses of sufentanil may be due to the different mechanisms of action at these concentrations. While at lower concentrations, sufentanil may not have reached the threshold for inducing cell proliferation inhibition, at higher concentrations, sufentanil may have exerted the inhibitory effect on cell proliferation *via* different pathways or mechanisms. In addition, other factors, such as differences in treatment duration, may have contributed to the observed differences.

Research into rocuronium mainly focuses on its effects on neuromuscular aspects (*Koo et al., 2020*; *Zhang et al., 2019*) and rarely on lung cancer cell function. We found that rocuronium inhibited lung tumor cell proliferation at regular and high-dose concentrations. We assume high concentrations of this drug significantly repressed tumor cell proliferation because they may be toxic to cells. Therefore, the growth rate of lung cancer cells decreased at a blood CB concentration of 23.5 μmol/L. This result suggests that rocuronium should be used at an appropriate dose during a surgical procedure, which is beneficial for the prognosis of patients with lung cancer (*Jiang et al., 2017*). Because the components of multi-drug anesthesia probably have additional potential targets, efficacy may be achieved through synergistic and dynamic interactions between them. Thus, the combined use of different anesthetic drugs may have an inhibitory effect on tumor cells during an operation. In clinical practice, anesthetics such as rocuronium, sufentanil, and propofol are often administered together to patients. Specific anesthetic agents and doses are applied in different types and locations of surgery (*Eger 2nd, 2001*). For example, *Lu & Xu (2006)* established in a retrospective study that patients with cancer undergoing an operation under total intravenous anesthesia had a better outcome than those under volatile inhalation anesthesia, suggesting that intravenous co-administration of anesthetic agents may contribute to killing tumor cells released into the circulation.

The OED was a central part of this study since it determines the level status of each factor and the interaction between, determining their optimal combination with a minimal number of sampling tests (*López-Cacho et al., 2012*; *Zhou et al., 2012*). We first used L9 ($3^3$) array to test the inhibitory effects of the three drugs at three concentrations on the proliferation of A-549 cells. We evaluated a total of nine combinations of the drugs and identified their optimal concentrations for the combined drug treatment. While all nine combinations inhibited the proliferation of human lung adenocarcinoma A-549 cells *versus*

control cells to varying degrees, the differences in the extent of inhibition were the most obvious at 24 h and when cells were exposed to the optimal drug concentrations. One possible explanation for the lack of significant differences at extended time points is that the inhibitory effects of the combined drugs on A-549 cell viability may have reached a plateau, resulting in similar levels of inhibition over time. Alternatively, other factors, such as cell cycle arrest or cellular adaptation, may have come into play at extended time points, reducing the observed differences.

This study showed that sufentanil exerted the most pronounced suppressive effect on lung cancer cells, followed by propofol and rocuronium in descending order. Interestingly, while the variance analysis revealed that only sufentanil could significantly inhibit cell proliferation, the combined drugs in optimal combination inhibited tumor growth better than either drug.

The first explanation for these results may stem from the heterogeneity and diversity of tumor cells within and between tumors, driving differences in the sensitivity of lung cancer cells to different narcotic drugs. The second reason is that tumor cells and their microenvironment interact in a complex and multifaceted manner to promote proliferation and metastasis (*Guan, 2015*; *Quail & Joyce, 2013*). Hence, any changes in the microenvironment in response to a narcotic treatment may impact tumor cell proliferation, migration, and invasion in culture. Although these mechanisms were not addressed in this study, we can suggest several from recent evidence that indicates tumor cell growth inhibition arises by blocking specific signaling pathways. For example, propofol inhibits A-549 cell growth, migration, and invasion by miR-372 downregulation (*Sun & Gao, 2018*). *Zheng et al. (2020)* found that propofol inhibits the growth of NSCLC cells and accelerates their apoptosis by regulating the miR-21/PTEN/AKT pathway *in vitro* and *in vivo*. Moreover, *Xing et al. (2018)* demonstrated that propofol inhibits the viability of A-549 cells and triggers their apoptosis *via* an ERK1/2-dependent pathway. Similarly, sufentanil may prevent lung cancer cells from proliferating and undergoing interstitial transition by obstructing the Wnt pathway (*Guan, Huang & Lin, 2022*).

Considering all these points, combinatory drug therapy may benefit from the cross-sensitization in tumor cells caused by the cross-modulation of molecular pathways targeted by a single drug (*Arroyo et al., 2020*). We speculate that the success of our 3-drug optimal combination in a clinical setting would depend on its ability to block/activate the signaling output of specific molecular pathways, which may exert synergistic effects on tumor suppression. Therefore, combining several single effective drugs can induce the same or greater extent of cell death or lower proliferation. These potential functions of the signaling pathway could exist simultaneously, or one may have a precedence role over the others. In addition, the effects of various factors on cell inhibition were directly described in the study while searching for the optimal combination conditions for inhibiting cell proliferation. However, since an individual factor in OED optimization is a multivariable combination, others may have a greater impact on cell inhibition.

Therefore, unlike single-variable optimization, an individual variable cannot simply reflect its changes in action. A reason for this may be the different trends observed in a plot

as a function of each factor and could explain why sufentanil had higher effectiveness than the other drugs in the three-drug combination.

A hallmark characteristic of cancer cells is their capability to migrate to nearby and distant tissues. This study revealed that the combined treatment with three anesthetics distinctly reduced the migratory and invasive abilities of A-549 cells, supporting the anti-migratory and anti-invasive role of the three-drug combination. In addition, the three-drug combination upregulated the expression of the apoptosis-related protein PARP1 in A-549 cells. These findings indirectly confirm that the three-drug combination has synergistic antitumor effectiveness in an *in vivo* lung cancer model.

Although this study offers valuable insights, it has some limitations. First, it was conducted *in vitro* using human lung adenocarcinoma A-549 cells, which may not fully recapitulate the complex tumor microenvironment and systemic effects *in vivo*. Therefore, while the orthogonal design allows the systematic optimization of multiple factors, it may not have the statistical power to identify the optimal conditions with high confidence. Moreover, because optimal drug dose, timing, and duration are essential for the effectiveness of the three-drug treatment, the orthogonal design may not accurately reflect the optimal doses and scheduling in clinical practice. Additionally, possible interactions between the three anesthetics should be considered; certain drugs affect the potency or toxicity of others when used in combination with implications for patient safety. Therefore, further preclinical and clinical studies are necessary to evaluate the safety and efficacy of the three-drug treatment before clinical use. Second, our study focused on the effects of propofol, sufentanil, and rocuronium on A-549 cell viability and proliferation but did not investigate the molecular mechanisms involved in the antitumor activity of these drugs. Hence, the underlying mechanisms of the antitumor effects exerted by the combined anesthetics demand future research for elucidation. Despite these limitations, our findings provide invaluable preliminary evidence of the antitumor activity of co-administered propofol, sufentanil, and rocuronium, warranting further preclinical and clinical studies to fully understand their therapeutic value in patients with cancer undergoing surgery.

## CONCLUSION

Here, we present a first comprehensive analysis of the factors influencing the proliferation of lung cancer A-549 cells, using experimentally single and combined applications of three clinically relevant intravenous anesthetic agents. Utilizing the OED method, we identified the critical factors affecting the growth of A-549 cells and defined optimal tri-drug combination conditions. Our experimental findings may serve as preliminary guidance on the compatibility of three anesthetic doses for patients with lung cancer undergoing surgical procedures. Importantly, our results represent the basis for future large-sample, randomized clinical trials.

### Funding

The authors received no funding for this work.

### Competing Interests

The authors declare there are no competing interests.

### Author Contributions

- Jing Tan conceived and designed the experiments, performed the experiments, prepared figures and/or tables, and approved the final draft.
- Lijun Wang performed the experiments, analyzed the data, authored or reviewed drafts of the article, and approved the final draft.
- Xuming Song performed the experiments, prepared figures and/or tables, and approved the final draft.
- Yijian Zhang analyzed the data, prepared figures and/or tables, and approved the final draft.
- Zhenghuan Song conceived and designed the experiments, authored or reviewed drafts of the article, and approved the final draft.
- Manlin Duan conceived and designed the experiments, authored or reviewed drafts of the article, and approved the final draft.

### Data Availability

Figshare: https://doi.org/10.6084/m9.figshare.22324006.

### Supplemental Information

Supplemental information for this article can be found online at http://dx.doi.org/10.7717/peerj.15672#supplemental-information.

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
