# Peer review of "Optimization of a tri-drug treatment against lung cancer using orthogonal design in preclinical studies"

_PeerJ, doi:10.7717/peerj.15672_

## Round 0.1 · original submission · Major Revisions

The language of the manuscript needs further polishing. In addition, attention should be paid to the uniform format of charts and references. In a chart, the standard deviation of the normally distributed data is essential.

Reviewer 1 ·

Basic reporting

1. Although the manuscript can be basically understood, English grammar still requires proofreading by fluent English speakers.
2. The citation and format of references basically comply with the standards of PeerJ magazine and a certain proportion of literature in the past three years.
3. In the Abstract, the author should clarify whether the purpose of this study is to explore the effects of anesthetics on postoperative outcomes in lung cancer patients or the role and potential mechanisms of combined administration of anesthetics in the treatment of lung tumors.
4. Please verify if the legend of the vertical axis is correct. In addition, the author should indicate in the caption what the groups in the figure represent, such as A1B1C1, A1B2C3, etc.

Experimental design

1. Authors' study demonstrates that clinical concentrations of propofol, sufentanil, and rocuronium can inhibit the proliferation and invasion of human lung adenocarcinoma A549 cells. Moreover, the combination of the three drugs at the optimized concentrations resulted in greater anti-tumor activity than any individual drug alone. These findings suggest that the co-administration of anesthetics may have potential therapeutic value in enhancing cancer outcomes after surgery. Further studies are needed to elucidate the underlying mechanisms and to validate these findings in clinical settings.Overall, the structure of the article is relatively clear and has certain publication value.
2. In materials and methods, the author indicated that “A viability assay was performed in which A549 cells were incubated with propofol (1.4, 4.2, and 12.6 umol/L), sufentanil (2/3, 2, and 6nmol/L), and rocuronium (2.61, 7.83, and 23.5 umol/L) for12, 24, 36 and 48 h.”. The author should explain why these concentrations of propofol, sufentanil and rocuronium were chosen for the experiment.
3. In order for readers to better understand the content of this study and improve the credibility of the article, the material methods need to be further detailed, such as Western blot analysis.

Validity of the findings

1. The description of the results is not clear and requires significant improvement, including the contents and the accuracy of the description corresponding to the Figures.
2. The author indicated that “Differences were most noticeable at 24 hours, and extended time points were not significant (Figure 1).” However, sufentanil, propofol and rocuronium respectively inhibited A549 cell viability in 36 or 48h after treatment. Therefore, what is the reason for the insignificant difference over time that requires further verification and analysis.
3. The author indicated that “When compared with the control group (1.00 ± 0.1), sufentanil 2/3 and 2 nmol/L treatment did not significantly affect cell proliferation. It was observed in A549 cells following 12, 24, and 48 h of sufentanil treated with 6 nmol/L (control, (1.00 ± 0.0); (0.57 ± 0.02); (0.59 ± 0.02); (0.66 ± 0.01) Figure 4A, B, D). Observations outlined above indicate that sufentanil inhibited A549 cell proliferation dose-dependently.” These results are not enough to draw such a conclusion. In addition, the author should explain why it could inhibit cell viability with 2nnmol sufentanil while could not inhibit cell viability with 6nnmol sufentanil.

Additional comments

The author should clearly explain the limitations of this study.

Reviewer 2 ·

Basic reporting

1.The manuscript writing needs to be polished as a whole.
2.This paper try to evaluate the effects of 3 anesthetic drugs on proliferation, and invasion of lung cancer A549 cells, The research intention has certain clinical value, and it is recommended to add more literature to indicate the motivation of this study.
3.The marking style of the bar chart in Figures should be unified, and it is recommended to refer to Figure 6C for other figures

Experimental design

1.Orthogonal design optimization provides preliminary information on the optimal dosage and exposure times for the tri-drug regimen. However, there are limitations to consider when extrapolating the results to human trials. Therefore, it is essential to use orthogonal design optimization in combination with other preclinical and clinical studies to assess the efficacy and safety of the tri-drug regimen fully. Although orthogonal design allows the systematic optimization of several factors simultaneously, the sample size may not have the statistical power required for identifying the optimized conditions with high confidence. Optimal drug dosage, timing, and duration are crucial for the tri-drug regimen's efficacy. Orthogonal design may not accurately reflect the optimal dosages and scheduling in clinical practice. In addition, although orthogonal design allows for the optimization of the tri-drug regimen's dosages and exposure times, it may not account for drug interactions between the three drugs. Certain drugs may affect the potency or toxicity of the other drugs when used in combination, which could have implications for patient safety.
2.The decrease in cell viability and the decrease in cell proliferation capacity are two different concepts. We suggest that the authors revise the relevant description, and if the authors insist that the combination of these three drugs inhibited the proliferation of A549 cells, then they should at least add an apoptosis experiment to demonstrate that the decrease in cell viability was not due to apoptosis.
3.Preclinical studies provide an initial assessment of a drug regimen's safety and efficacy, but it does not guarantee that the results will translate accurately in human trials. Factors such as genetic variability and individual differences in drug absorption, metabolism, and activation could affect the tri-drug regimen's efficacy in clinical settings. Therefore, to ensure the effectiveness of this study, animal experiments should be added to this study.

Validity of the findings

1.Please indicate the repetition of the experiment in Figure legends, and all Control and NC groups do not have SD values.
2.The images in Figure 6 require the addition of rulers, and the WB images require data statistics.
3.Suggest re checking the writing in the results section for easier understanding.

Additional comments

Introduction section more clinical significance should be fully mentioned that propofol, sufentanil or rocu are often used together in surgical intervention, which is the main value of this study.

---

## Round 0.2 · accepted · Accept

The authors have addressed the reviewers' concerns.

Reviewer 1 ·

Basic reporting

no comment

Experimental design

no comment

Validity of the findings

no comment

Additional comments

The author's revisions and responses can be generally acknowledged,I Agree to the acceptance of the manuscript and hope that the author will continue to conduct in-depth research in the future.

Reviewer 2 ·

Basic reporting

The author has made efforts to revise and respond, and I think their revisions can be recognized.

Experimental design

no comment

Validity of the findings

no comment